health and disease and epidemiology

COVID-19, Europe, epidemiology, modelling

**Author for correspondence:**
Christopher Dye
e-mail: christopher.dye@zoo.ox.ac.uk

# The scale and dynamics of COVID-19 epidemics across Europe

Christopher Dye[1], Russell C. H. Cheng[2], John S. Dagpunar[2] and Brian G. Williams[3]

[1]Department of Zoology, University of Oxford, Oxford, UK
[2]Department of Mathematical Sciences, University of Southampton, Southampton, UK
[3]South African Centre for Epidemiological Modelling and Analysis (SACEMA), Stellenbosch University, Stellenbosch, South Africa

CD, 0000-0002-2957-1793; RCHC, 0000-0001-7836-7820;
JSD, 0000-0001-7197-8231; BGW, 0000-0002-3174-4876

The number of COVID-19 deaths reported from European countries has varied more than 100-fold. In terms of coronavirus transmission, the relatively low death rates in some countries could be due to low intrinsic (e.g. low population density) or imposed contact rates (e.g. non-pharmaceutical interventions) among individuals, or because fewer people were exposed or susceptible to infection (e.g. smaller populations). Here, we develop a flexible empirical model (skew-logistic) to distinguish among these possibilities. We find that countries reporting fewer deaths did not generally have intrinsically lower rates of transmission and epidemic growth, and flatter epidemic curves. Rather, countries with fewer deaths locked down earlier, had shorter epidemics that peaked sooner and smaller populations. Consequently, as lockdowns were eased, we expected, and duly observed, a resurgence of COVID-19 across Europe.

The total number of COVID-19 deaths reported by European countries up to 31 July 2020 varied more than 100-fold, from approximately 100 in Croatia to more than 45 000 in the UK. In terms of the dynamics of coronavirus transmission, there are broadly three possible reasons why a country might suffer relatively few deaths. The first is that the transmission rate of the coronavirus, SARS CoV-2, is intrinsically lower in some countries, for example, because infectious and susceptible individuals come into contact less frequently in less dense populations. This would be reflected in a relatively low value of the basic case reproduction number, $R_0$. Figure 1$a$ shows how $R_0$ changes the shape and scale of an epidemic, aided by a dynamic SEIR epidemiological model (electronic supplementary material), and with reference to deaths reported from Germany. Given a basic case reproduction number of $R_0 = 3$, the number

**Figure 1.** An SEIR transmission model (solid lines) illustrates the mechanisms that determine the scale and dynamics of COVID-19 epidemics, here described by the number of deaths reported each week. (a) Lower values of the time-invariant, basic case reproduction number ($R_0 = 6$, 3 or 2, adjusted, for example, by social mixing) mitigate transmission and flatten the epidemic curve. With lower $R_0$, the maximum number of deaths per week is reduced and delayed, generating epidemics that are smaller and longer. (b) After the initial period of epidemic growth (governed by $R_0$), the decline of the effective case reproduction number, $R_t \leq R_0$, is accelerated by limiting the size of the susceptible population (1.5, 1.0 or 0.5 million people), generating epidemics that are smaller and shorter, peaking earlier. The epidemics in (a) or (b) could also be replicated by reducing $R_0$ to $R_{0c}$ (with control measures) before or during the epidemic, for example, through non-pharmaceutical interventions (NPI). (a) and (b) also show weekly reported deaths in Germany (points) described by the SEIR model and, for comparison, the skew-logistic model (S–L, dotted line; figure 2). The SEIR model and data [1] are described further in the electronic supplementary material.

of deaths reached a maximum of approximately 1600 in the week of 16 March, and an estimated total of 9300 people died from COVID-19 by the end of July (figure 1a). If $R_0$ had been greater at the outset ($R_0 = 6$), the epidemic would have been larger (9800 deaths) and shorter, growing faster and peaking sooner. If $R_0$ had been lower ($R_0 = 2$), the epidemic would have been smaller (7900 deaths) and longer, growing more slowly with a delayed peak. A lower value of $R_0$ mitigates transmission and flattens the epidemic curve, protecting both health and health services [2–8].

The second possibility is that some countries have suffered fewer deaths because, after the initial period of maximum epidemic growth (governed by $R_0$), the effective case reproduction number ($R_t \leq R_0$) declined relatively quickly through time in smaller populations of susceptibles, depleted by the build-up of herd immunity [9] (figure 1b). This is a simple representation of heterogeneous exposure or susceptibility to infection which, compared with homogeneous exposure or susceptibility, would lower the fractions of people who become infected, ill and die [10,11]. Countries with fewer people exposed or susceptible to infection are expected to have smaller and shorter COVID-19 epidemics (figure 1b).

The third possible reason for fewer deaths is that the basic reproduction number, $R_0$, was reduced to a lower value, $R_{0c}$, by control methods imposed before or during the epidemic. These are predominantly 'non-pharmaceutical interventions' (NPI), which include personal (physical distancing, isolation, quarantine, hand hygiene, face covering), environmental (surface cleaning and ventilation) and social (travel restrictions, school and workplace closures, restriction on mass gatherings) methods of

preventing contact and transmission between infectious and susceptible individuals. In this model of epidemic control, the case reproduction number is reduced through time, not because susceptibles are progressively depleted in an exposed subpopulation, but because they can no longer be contacted by infectious cases [4,12,13]. A change from $R_0$ to $R_{0c}$ could replicate the effects seen in either figure 1a or b, dependent on the timing and magnitude of the change, but independent of population size (figure 1b). A consequence of this model of epidemic control is that, if NPI are relaxed in a population where many people are still susceptible to infection, we expect a resurgence of COVID-19.

Which of these three possibilities explains the variation in COVID-19 deaths among European countries? In this study, rather than presuming which mechanism of epidemic control is more likely *a priori*, we began by developing a flexible, empirical model (skew-logistic) to characterize the scale and dynamics of COVID-19 epidemics in European countries (electronic supplementary material). Also, an extra motive for choosing an empirical model is that the SEIR model above cannot accurately describe European COVID-19 epidemics when constrained by biologically plausible parameter values, notably the slow rates of decline in the asymmetric epidemics. Nevertheless, the skew-logistic model also has drawbacks. For example, the epidemics in Albania, Bulgaria, Iceland, Ireland, Serbia and Slovakia could not be described by the skew-logistic, either because there were too few data (e.g. Iceland) or because the form of the epidemic was not skew-logistic (Bulgaria).

Unconstrained by biological assumptions, and agnostic to the mechanisms of epidemic control (figure 1), the skew-logistic model allowed us to measure, independently of each other, the key components of an epidemic: the rate of epidemic growth, the size of the epidemic peak (maximum number of deaths per week), the period or duration of growth (estimated time from one death to the maximum number of weekly deaths) and the rate of decline. We then asked which of these elements best explains the differences among European countries, and by what mechanism (figure 1).

The skew-logistic model gives an excellent description of COVID-19 epidemics in 24 European countries (figures 1 and 2; electronic supplementary material). The average initial growth rate of epidemics was $0.28 \, \mathrm{day}^{-1}$ (95% CI $\pm 0.006 \, \mathrm{day}^{-1}$), doubling time 2.5 days), the average period of epidemic growth was 37 days (95% CI $\pm 4.0$ days), and the average rate of decline was $-0.05 \, \mathrm{day}^{-1}$ (95% CI $\pm 0.001 \, \mathrm{day}^{-1}$, halving time 14.1 days). For each of the 24 countries, maximum-likelihood estimates for parameters of the skew-logistic are tabulated in the electronic supplementary material.

The skew-logistic model shows how the number of COVID-19 deaths was associated with these characteristics of the epidemics and the size of national populations. In general, countries reporting fewer COVID-19 deaths had fewer inhabitants (Student's $t = 6.83$, $p < 0.001$; figure 3a). In countries reporting fewer deaths, epidemics neither grew more slowly (figure 3b) nor declined more quickly (figure 3c). Rather, epidemics that caused fewer deaths were curtailed; they grew for shorter periods of time ($t = 3.91$, $p < 0.001$; figure 3d). And shorter epidemics were smaller epidemics, even though shorter epidemics tended to increase more quickly ($t = 3.95$, $p < 0.001$; figure 3e).

The product of the rate (*g*) and period of growth (*d*) defines net epidemic growth—how much an outbreak expands in size during the growth period. Net growth, measured by $G = e^{gd}$, accounted for 83% of the inter-country variation in the number of reported deaths ($t = 12.4$, $p < 0.001$; figure 3f). Although such a relationship is expected in principle, it is surprising that theory is upheld so faithfully in data collected in different ways across 24 diverse European countries.

Drawing together the data across these 24 countries (figure 3), multiple regression analysis shows that both population size (*N*) and net growth (*G*) were strongly and independently associated with the total number of deaths (*D*), such that $\ln(D) = 0.31 \ln(N) + 0.67 \ln(G)$. The number of deaths therefore increased as a power function (less than proportionally) with population size ($D \propto N^{0.31}$, $0.31 < 1$, $t = 4.1$, $p < 0.001$) and net growth ($D \propto G^{0.67}$; $0.67 > 0$, $t = 13.9$, $p < 0.001$; overall $R^2 = 0.83$) and, given the definition of *G*, exponentially with epidemic growth rate and period ($D \propto e^{0.67gd}$). The finding that $D \propto N^{0.31}$ indicates that the number of inhabitants of a country is not simply an epidemic scaling (proportionality) factor with effects that can be captured as deaths per capita, as implied in some presentations of the data [14,15].

Among competing explanations for the large inter-country variation in reported deaths, these data are consistent with the view that there were systematic national differences in the effective case reproduction number ($R_t$ and/or $R_{0c}$, figure 1b) but not the basic case reproduction number ($R_0$, figure 1a). What, then, were the relative roles of $R_t$ and $R_{0c}$ as determinants of the numbers of deaths?

Because the period of epidemic growth (figure 3d) and net growth (figure 3f) accounted for much of the inter-country variation in reported deaths, the timing of interventions is expected to influence the total number of deaths, moderated by population size (figure 3a). One summary measure of NPI is the COVID-19 Containment and Health Index (CHI), a composite metric based on 11 indicators

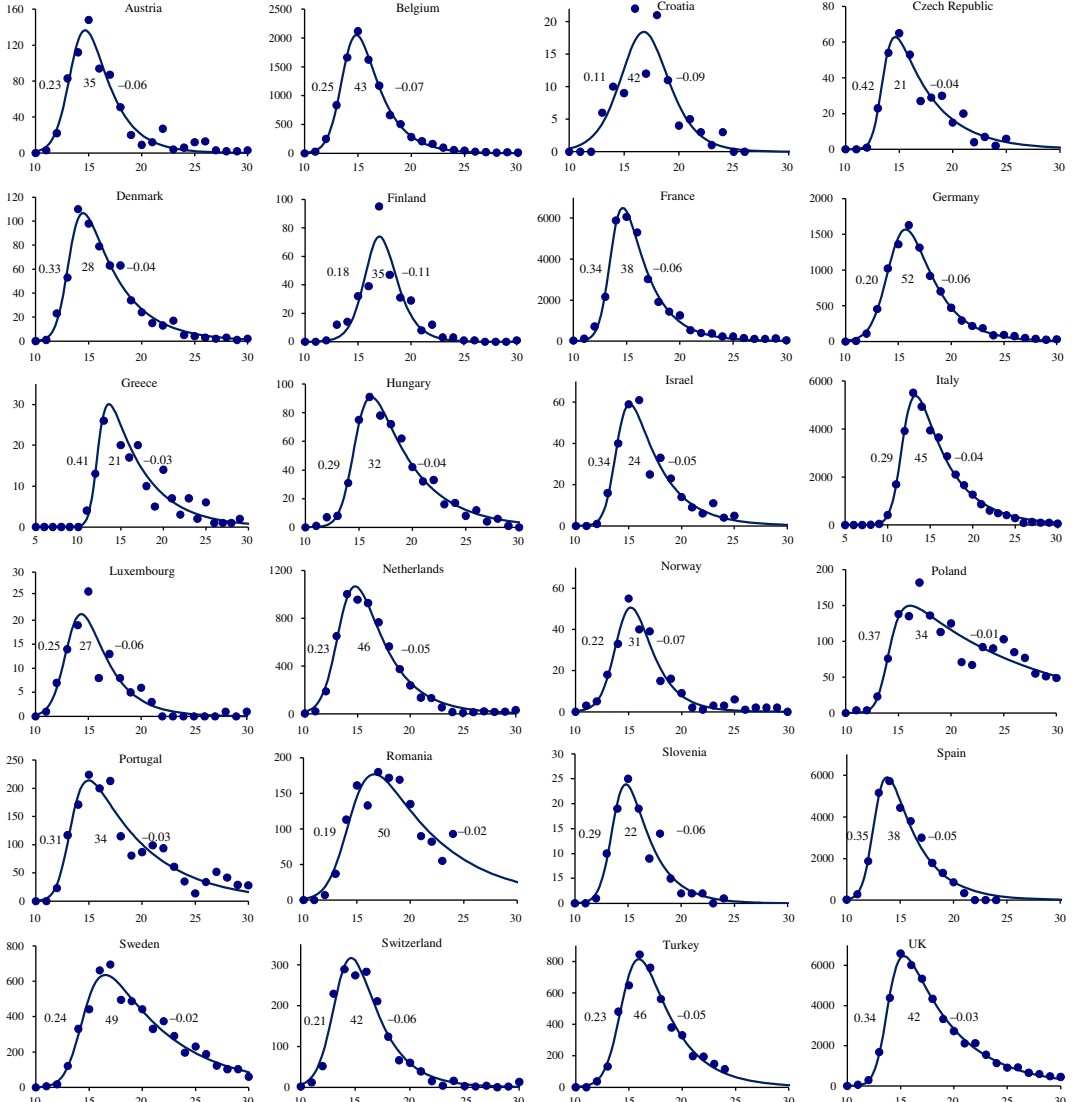

**Figure 2.** Epidemic curves for 24 European countries based on the number of deaths (points, vertical axis) reported each week (horizontal axis), described by the skew-logistic model (line) which is used to calculate the epidemic growth rate, the period of epidemic growth and the rate of decline in each country (inset figures, left to right). Although deaths are reported daily from each country, and growth rates and periods are given in days, the model is fitted to the weekly numbers of deaths in order to sum over the typical 7-day reporting cycle (electronic supplementary material). Data source [1].

including containment and closure policies (schools, workplaces, travel bans) and health system policies (information, testing, contact tracing), with a score for each country varying between 0 and 100% (electronic supplementary material) [16].

Figure 4a shows that the total number of deaths (D) reported from 30 countries (the 24 in figures 2 and 3, plus Albania, Bulgaria, Iceland, Ireland, Serbia and Slovakia) was closely associated with the number predicted on the basis of population size (N) and the cumulative number of deaths that had been reported by the date of lockdown ($D_L$), as judged from the CHI (different interpretations of the time of lockdown give similar results; electronic supplementary materials). In figure 4a (similar to figure 3f), the number of deaths increased as a power function of population size ($D \propto N^{0.49}$, $0.49 < 1$, $t = 5.3$, $p < 0.001$) and with the number of deaths at lockdown ($D \propto D_L^{0.67}$; $0.67 > 0$, $t = 3.4$, $p < 0.002$; overall $R^2 = 0.84$). Thus, to a good approximation, the number of deaths in each country varied in proportion to $N^{1/2}$ and $D_L^{2/3}$.

Considering all 30 countries together, lockdown evidently stopped and reversed the increase in COVID-19 deaths in European countries (figure 4b). Most of the 30 countries investigated here locked down within a short period in March and the CHI surpassed 70% on average by 16 March.

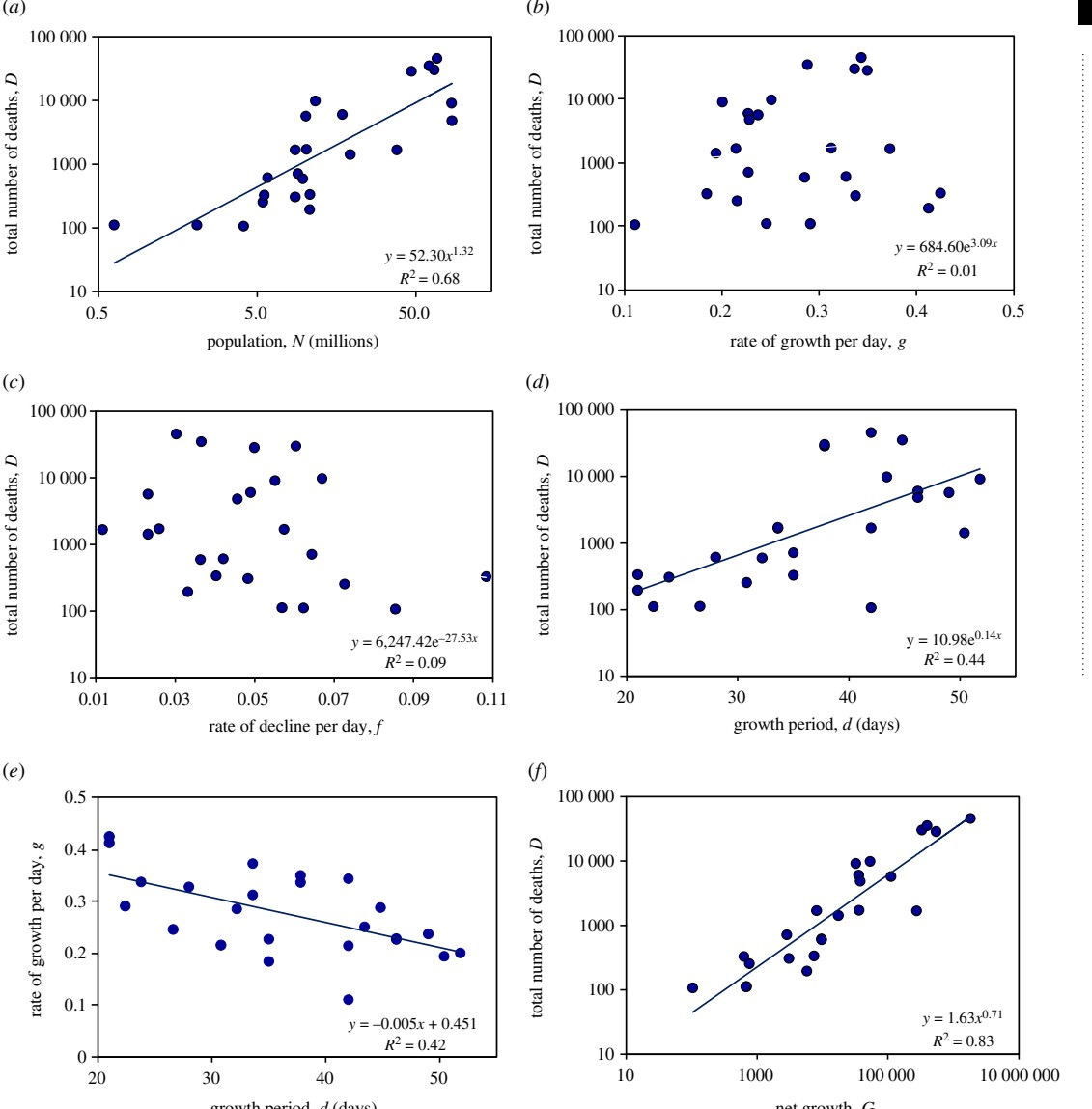

**Figure 3.** Determinants of the total number of reported COVID-19 deaths in each of 24 European countries, using estimates derived from the skew-logistic model (figure 2). (*a*) The number of deaths increased with national population size. The number of deaths varied independently of (*b*) the rate of epidemic growth and (*c*) decline. But (*d*) more deaths were reported when epidemics were allowed to grow for longer, even though (*e*) longer periods of growth were associated with lower epidemic growth rates. (*f*) The number of deaths was strongly associated with growth rate and period combined as net growth, $G = e^{gd}$. Each panel gives the regression equation (regression line shown if statistically significant) and the fraction of the variation explained by each independent variable ($R^2$). Data source [1].

Lockdown was followed by a progressive fall in the daily growth rate of deaths (daily change in the 7-day running mean), so that deaths were no longer increasing (negative growth) by the second week in April.

Just as the lockdown appears to have prevented the spread of COVID-19, the easing of restrictions preceded a resurgence. The average value of the CHI peaked at 75% on 12 April and was held above 70% for 46 days. As lockdowns were eased across Europe, more variably among countries than they were imposed (widening confidence intervals in figure 4*b*), the decline in deaths stalled. The CHI fell to 50% on average by 15 July by which time the daily growth rate of deaths was once again approaching zero (figure 4*b*). From June onwards, the daily growth rate was often above zero, consistent with reports of renewed outbreaks of COVID-19 across the European region, and reinforcing the view that the previous rise in deaths had been constrained, at least in part, by lockdown (figure 4*a*).

In conclusion, this investigation differs from others in using an empirical model (skew-logistic, devised for this study), to explore the factors that affect the size of COVID-19 epidemics (measured in

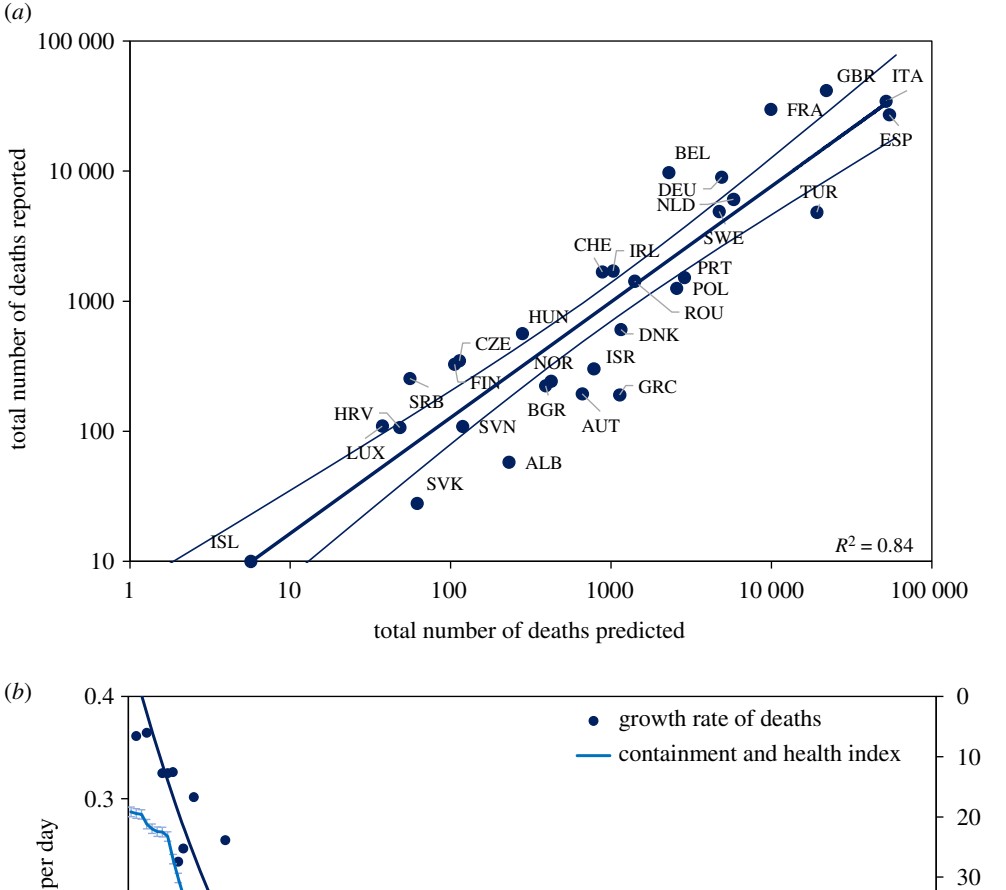

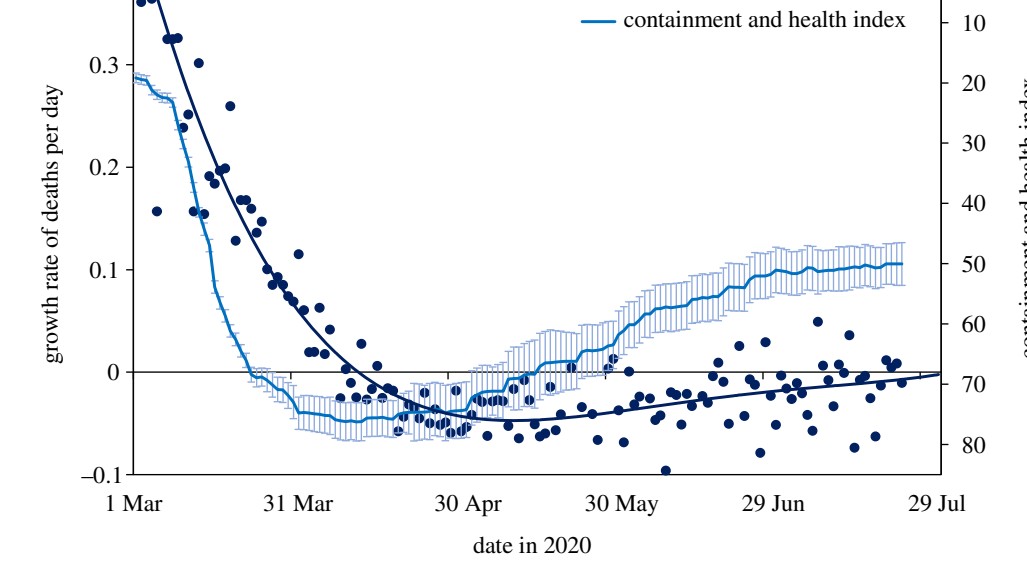

**Figure 4.** (a) Total numbers of COVID-19 deaths (D) reported from 30 European countries (the 24 in figures 2 and 3 plus Albania, Bulgaria, Iceland, Ireland, Serbia and Slovakia) in relation to the number of deaths predicted from population size (N) and the number reported at the time of lockdown ($D_L$). Errors on the regression line mark 95% CI. (b) The imposition and release of lockdown across 30 countries, as measured by the mean value of the Containment and Health Index (light blue, error bars 95% CI, reverse scale on y-axis), which was followed by the changing daily growth rate of deaths (dark blue points, fitted polynomial line of order 4). Further information and analysis, including full country names, are in the electronic supplementary material. Data sources [1,16].

terms of reported deaths) across Europe. We find that countries reporting fewer deaths did not generally have lower rates of transmission and epidemic growth; that is, they did not have flatter epidemic curves, driven by lower values of $R_0$ (contrary to figure 1a). Rather, countries with fewer deaths were those that locked down earlier (having reduced $R_0$ to $R_{0c}$) generating shorter epidemics that peaked sooner (figure 1b). Similar effects of lockdown have been inferred from previous analyses of COVID-19 in European [4,14] and other countries, including China [17–21].

Fewer deaths were also reported by countries with fewer inhabitants, for reasons that are not yet clear. If lockdown and NPI were the sole mechanism of epidemic control, we would expect no association between the number of deaths and population size. One possible explanation is that

smaller countries had fewer introduced infections from which national epidemics grew. Another is that only a subset of any national population is exposed or susceptible to infection and illness [10,11], and that subpopulation is smaller in countries with fewer inhabitants. The distribution of coronavirus is highly heterogeneous in every country, as any map of COVID-19 subnational distribution shows [22]. This latter hypothesis implies that epidemics have been controlled, in part, by the local (subnational) depletion of susceptibles and the build-up of herd immunity (changes in $R_t$ falling below $R_0$; figure 1$b$).

This analysis, like previous studies [4,14,17], exposes the perils of delayed action during an epidemic with an average doubling time as short as 3 days. At this rate of growth, the daily death toll would have increased by a factor of 10 within 9 days; this is about the same as the average time delay from one death to the time of lockdown, which was 9.0 (95% CI ± 2.6) days. In effect, a COVID-19 epidemic in a typical European country would have expanded more than 10-fold within the average time it took to impose lockdown.

The need for speed is expected to apply to resurgent epidemics too. The strong link between COVID-19 deaths and the time of lockdown implies that only a small fraction of Europe's population was exposed to infection in the first epidemic wave, a view reinforced by serological surveys that generally report less than a 10% national prevalence of antibody to SARS CoV-2 [14,23–25]. Consequently, like other investigators [2,4,12,15,26], we expect that further COVID-19 outbreaks will continue to threaten large numbers of susceptible people across Europe. The apparent effectiveness of lockdowns (figure 4$a$), and the penalty for releasing them (figure 4$b$), underline the central dilemma facing European countries: how to maintain the beneficial effects of physical distancing without reinstating full lockdowns across the continent.

As a comparative analysis of European countries—which has identified a strong association between COVID-19 deaths, the timing of lockdown and population size—this study says nothing about the importance of other factors that could affect COVID-19 epidemiology. Among these factors are environmental risks such as air pollution [27], underlying health conditions [28] including obesity [29], possible protective cross-immunity to other coronaviruses [30,31], ethnicity and occupation [32,33], demography and age structure [34,35], and methods of treatment and clinical care that could lower case fatality [36,37]. Whether any of these factors can help to explain the epidemiological differences between countries, with a view to finding better ways of containing COVID-19 across Europe, remains an open question.

Data accessibility. All data are available in the main text, in the electronic supplementary material and in the sources cited. Software to fit the skew-logistic model is available from the authors and from the Dryad Digital Repository cited in the electronic supplementary material.

Authors' contributions. The authors jointly formulated the research goals and aims, designed the study and compiled the data. R.C.H.C., J.S.D. and B.G.W. developed the mathematical and statistical models and estimation routines. C.D. wrote the first draft of the paper, which was finalized with contributions from all authors.

Competing interests. The authors declare no competing interests.

Funding. C.D. was supported by a Visiting Fellowship at the Oxford Martin School.

Acknowledgements. We thank Charles Godfray, Moritz Kraemer and Oliver Pybus for helpful suggestions.

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
