## [Reviewer comments · Royal Society Open Science]

Review History

RSOS-201726.R0 (Original submission)

Review form: Reviewer 1

Is the manuscript scientifically sound in its present form?

Yes

Are the interpretations and conclusions justified by the results?

Yes

Is the language acceptable?

Yes

Do you have any ethical concerns with this paper?

No

Have you any concerns about statistical analyses in this paper?

No

Recommendation?

Accept with minor revision (please list in comments)

Comments to the Author(s)

This study describes an empirical model (skew-logistic model) developed to measure key components of epidemics, e.g. growth, size, duration of growth and the rate of decline. The model was fitted to COVID-19 deaths reported from 24 European countries and the estimates used to examine three possible explanations for the variation in COVID-19 deaths among European countries: variation in basic reproduction number (R_0), decline in effective case reproduction number and reduction of R_0 to a lower value by “control methods imposed before or during the epidemic”.

The skew-logistic model approach proposed is a straightforward alternative to model epidemics curves that cannot be adequately described by an SEIR model, e.g. as the authors note “The SEIR model above cannot accurately describe European COVID-19 epidemics when constrained by biologically plausible parameter values, notably the slow rates of decline in the asymmetric epidemics.” Regression models were then used to examine the relationship between total number of deaths and estimates of the skew-logistic model parameters for each country and provide evidence for the proposed explanations for the variation in deaths, e.g. strong associations between duration of epidemic growth and net growth and death indicate that “timing of interventions is expected to influence the total number of deaths” and supporting an exploration of the impact of lockdown on deaths.

Limitations of the analysis are mentioned in the discussion (e.g. impact of factors, such as, co-morbidities, demography and age structure etc. were not considered) and the sensitivity of the results to the method used to determine date of lockdown was explored (e.g. Containment and Health Index versus Stringency Index). Overall, this study describes a nice approach to modelling epidemic curves that can identify key drivers of variation in COVID19 deaths among countries.

I have the following minor comments:

- Missing +/- for some confidence intervals (e.g. see page 3, line 7)
- I assume the “t” statistics in the main text (e.g. page 3, line 15) are Wald test statistics?
- May want to consider including these points from the Supplementary Materials in the main text
 - o “The SEIR model above cannot accurately describe European COVID-19 epidemics when constrained by biologically plausible parameter values, notably the slow rates of decline in the asymmetric epidemics.”
 - o “Albania, Bulgaria, Iceland, Ireland, Serbia and Slovakia for which epidemics could not be described by the skew-logistic, either because there were too few data (e.g. Iceland), or the data did not describe a standard epidemic curve (Bulgaria).”

Review form: Reviewer 2

Is the manuscript scientifically sound in its present form?

Yes

Are the interpretations and conclusions justified by the results?

Yes

Is the language acceptable?

Yes

Do you have any ethical concerns with this paper?

No

Have you any concerns about statistical analyses in this paper?

No

Recommendation?

Accept with minor revision (please list in comments)

Comments to the Author(s)

This manuscript presents a robust statistical analysis of the shape of epidemic during the first wave of the Covid-19 pandemic in Europe. The main strength (and limitation) of the study is using a descriptive empirical curve to estimate three key characteristics of epidemic curves - the exponential growth rate, timing and size of peak. With reference to a mechanical compartmental SEIR model they use the variation in these three quantities to conclude that the majority of variation between European countries is associated with the timing of imposition of lockdown measures.

This is a careful thorough descriptive analysis of public data sets that provides insight into the variability of outcomes during the first wave and has clear relevance for informing action during the second wave.

My only concern is with the comparison between the performance of the standard compartmental SEIR model and the skew logistic. Describing the skew logistic curves as "generally better" (Page 4, Line 3) is both vague and misleading given the well known effect of realistic infectious and latent period distributions on the rate of exponential growth, timing and peak size of epidemics (e.g. Wearing et al. <https://doi.org/10.1371/journal.pmed.0020174>). Uncoupling the relationship between exponential growth and peak to be free parameters, it is not surprising the skew logistic can give a better qualitative fit. Several independent estimates of the serial interval distribution for Covid-19 are now available (<https://doi.org/10.1056/NEJMoa2001316>, <https://doi.org/10.1016/j.ijid.2020.02.060>, <https://doi.org/10.3201/eid2606.200357>) which suggest the variability in the serial interval is less than exponential. Reference to a more realistic model with gamma-distributed infectious and latent periods would be a more meaningful comparison. However, I suspect this would be difficult to achieve Excel and strays from the main results of the paper. These caveats - and the point that the SEIR model is being used only as an illustration - should be acknowledged though.

Minor comments

Page 3, Line 20 The authors make the distinction through adding a subscript "C" but I find the notion of the basic reproduction ratio changing dynamically over the course of an epidemic problematic given that the standard definition specifically concerns the "first" generation. I would consider R_0^C to be an effective reproduction ratio in the presence of controls and reserve R_0 for the fully susceptible population in the absence of controls - but admit there is far from a consensus on this!

Decision letter (RSOS-201726.R0)

Dear Professor Dye,

On behalf of the Editors, we are pleased to inform you that your Manuscript RSOS-201726 "The scale and dynamics of COVID-19 epidemics across Europe" has been accepted for publication in Royal Society Open Science subject to minor revision in accordance with the referees' reports. Please find the referees' comments along with any feedback from the Editors below my signature.

Please submit your revised manuscript and required files (see below) no later than 7 days from today's (ie 11-Nov-2020) date. Note: the ScholarOne system will 'lock' if submission of the revision is attempted 7 or more days after the deadline. If you do not think you will be able to meet this deadline please contact the editorial office immediately.

on behalf of Professor Joshua Ross (Associate Editor) and Mark Chaplain (Subject Editor)
openscience@royalsociety.org

Reviewer comments to Author:

Reviewer: 1
Comments to the Author(s)

This study describes an empirical model (skew-logistic model) developed to measure key components of epidemics, e.g. growth, size, duration of growth and the rate of decline. The model was fitted to COVID-19 deaths reported from 24 European countries and the estimates used to examine three possible explanations for the variation in COVID-19 deaths among European countries: variation in basic reproduction number (R_0), decline in effective case reproduction number and reduction of R_0 to a lower value by "control methods imposed before or during the epidemic".

The skew-logistic model approach proposed is a straightforward alternative to model epidemics curves that cannot be adequately described by an SEIR model, e.g. as the authors note "The SEIR model above cannot accurately describe European COVID-19 epidemics when constrained by biologically plausible parameter values, notably the slow rates of decline in the asymmetric epidemics." Regression models were then used to examine the relationship between total number of deaths and estimates of the skew-logistic model parameters for each country and provide evidence for the proposed explanations for the variation in deaths, e.g. strong associations between duration of epidemic growth and net growth and death indicate that "timing of interventions is

expected to influence the total number of deaths” and supporting an exploration of the impact of lockdown on deaths.

Limitations of the analysis are mentioned in the discussion (e.g. impact of factors, such as, co-morbidities, demography and age structure etc. were not considered) and the sensitivity of the results to the method used to determine date of lockdown was explored (e.g. Containment and Health Index versus Stringency Index). Overall, this study describes a nice approach to modelling epidemic curves that can identify key drivers of variation in COVID19 deaths among countries.

I have the following minor comments:

- Missing +/- for some confidence intervals (e.g. see page 3, line 7)
- I assume the “t” statistics in the main text (e.g. page 3, line 15) are Wald test statistics?
- May want to consider including these points from the Supplementary Materials in the main text
 - o “The SEIR model above cannot accurately describe European COVID-19 epidemics when constrained by biologically plausible parameter values, notably the slow rates of decline in the asymmetric epidemics.”
 - o “Albania, Bulgaria, Iceland, Ireland, Serbia and Slovakia for which epidemics could not be described by the skew-logistic, either because there were too few data (e.g. Iceland), or the data did not describe a standard epidemic curve (Bulgaria).”

Reviewer: 2

Comments to the Author(s)

This manuscript presents a robust statistical analysis of the shape of epidemic during the first wave of the Covid-19 pandemic in Europe. The main strength (and limitation) of the study is using a descriptive empirical curve to estimate three key characteristics of epidemic curves - the exponential growth rate, timing and size of peak. With reference to a mechanical compartmental SEIR model they use the variation in these three quantities to conclude that the majority of variation between European countries is associated with the timing of imposition of lockdown measures.

This is a careful thorough descriptive analysis of public data sets that provides insight into the variability of outcomes during the first wave and has clear relevance for informing action during the second wave.

My only concern is with the comparison between the performance of the standard compartmental SEIR model and the skew logistic. Describing the skew logistic curves as “generally better” (Page 4, Line 3) is both vague and misleading given the well known effect of realistic infectious and latent period distributions on the rate of exponential growth, timing and peak size of epidemics (e.g. Wearing et al. <https://doi.org/10.1371/journal.pmed.0020174>). Uncoupling the relationship between exponential growth and peak to be free parameters, it is not surprising the skew logistic can give a better qualitative fit. Several independent estimates of the serial interval distribution for Covid-19 are now available (<https://doi.org/10.1056/NEJMoa2001316>, <https://doi.org/10.1016/j.ijid.2020.02.060>, <https://doi.org/10.3201/eid2606.200357>) which suggest the variability in the serial interval is less than exponential. Reference to a more realistic model with gamma-distributed infectious and latent periods would be a more meaningful comparison. However, I suspect this would be difficult to achieve Excel and strays from the main results of the paper. These caveats - and the point that the SEIR model is being used only as an illustration - should be acknowledged though.

Minor comments

Page 3, Line 20 The authors make the distinction through adding a subscript “C” but I find the notion of the basic reproduction ratio changing dynamically over the course of an epidemic problematic given that the standard definition specifically concerns the “first” generation. I would

consider R_0^C to be an effective reproduction ratio in the presence of controls and reserve R_0 for the fully susceptible population in the absence of controls - but admit there is far from a consensus on this!

===PREPARING YOUR MANUSCRIPT===

===PREPARING YOUR REVISION IN SCHOLARONE===

Author's Response to Decision Letter for (RSOS-201726.R0)

See Appendix A.

Decision letter (RSOS-201726.R1)

Dear Professor Dye,

It is a pleasure to accept your manuscript entitled "The scale and dynamics of COVID-19 epidemics across Europe" in its current form for publication in Royal Society Open Science.

COVID-19 rapid publication process:

We are taking steps to expedite the publication of research relevant to the pandemic. If you wish, you can opt to have your paper published as soon as it is ready, rather than waiting for it to be published the scheduled Wednesday.

This means your paper will not be included in the weekly media round-up which the Society sends to journalists ahead of publication. However, it will still appear in the COVID-19 Publishing Collection which journalists will be directed to each week (<https://royalsocietypublishing.org/topic/special-collections/novel-coronavirus-outbreak>).

If you wish to have your paper considered for immediate publication, or to discuss further, please notify openscience_proofs@royalsociety.org and press@royalsociety.org when you respond to this email.

on behalf of Professor Joshua Ross (Associate Editor) and Mark Chaplain (Subject Editor)
openscience@royalsociety.org

Appendix A

Response to reviewers and editors

Thanks to both reviewers for their helpful suggestions. Our replies to their few remarks (abbreviated below to the essentials) are at ***, and the changes are tracked in the main text and in the supplementary materials.

In addition to the replies below we have also now included reference to the Dryad Digital Database, and put the references in the Royal Society's Vancouver style, with doi for journal papers.

Reviewer comments to Author:

Reviewer: 1

Comments to the Author(s)

I have the following minor comments:

- Missing +/- for some confidence intervals (e.g. see page 3, line 7)

*** Fixed

- I assume the "t" statistics in the main text (e.g. page 3, line 15) are Wald test statistics?

*** It is Student's t-test; we have now made that clear

- May want to consider including these points from the Supplementary Materials in the main text
 - o "The SEIR model above cannot accurately describe European COVID-19 epidemics when constrained by biologically plausible parameter values, notably the slow rates of decline in the asymmetric epidemics."
 - o "Albania, Bulgaria, Iceland, Ireland, Serbia and Slovakia for which epidemics could not be described by the skew-logistic, either because there were too few data (e.g. Iceland), or the data did not describe a standard epidemic curve (Bulgaria)."

*** Both points have been included in the main text - thanks

Reviewer: 2

Comments to the Author(s)

My only concern is with the comparison between the performance of the standard compartmental SEIR model and the skew logistic. Describing the skew logistic curves as "generally better" (Page 4, Line 3) is both vague and misleading given the well known effect of realistic infectious and latent period distributions on the rate of exponential growth, timing and peak size of epidemics (e.g. Wearing et al. <https://doi.org/10.1371/journal.pmed.0020174>). Uncoupling the relationship between exponential growth and peak to be free parameters, it is not surprising the skew logistic can give a better qualitative fit. Several independent estimates of the serial interval distribution for Covid-19 are now available (<https://doi.org/10.1056/NEJMoa2001316>, <https://doi.org/10.1016/j.ijid.2020.02.060>, <https://doi.org/10.3201/eid2606.200357>) which suggest the variability in the serial interval is less than exponential. Reference to a more realistic model with gamma-distributed infectious and latent periods would be a more meaningful comparison. However, I suspect this would be difficult to achieve Excel and

strays from the main results of the paper. These caveats - and the point that the SEIR model is being used only as an illustration - should be acknowledged though.

*** Point taken about the imprecise phrase "generally better", so we have deleted that comparison, which is not needed anyway at that point in the text.

*** The point about non-exponential delay distributions is also valuable, so we have made ref in the Supplementary Materials to Wearing et al and also to the recent Royal Society SET-C paper on reproduction numbers that also covers this point.

*** The third point about SEIR being used for illustration has also been included in the Supplementary Materials

Minor comments

Page 3, Line 20 The authors make the distinction through adding a subscript "C" but I find the notion of the basic reproduction ratio changing dynamically over the course of an epidemic problematic given that the standard definition specifically concerns the "first" generation. I would consider R_0^C to be an effective reproduction ratio in the presence of controls and reserve R_0 for the fully susceptible population in the absence of controls - but admit there is far from a consensus on this!

*** We wanted to distinguish R_0^C (the modification of R_0 by control measures) from R_t ($< R_0$), which is the time-varying case reproduction number as infection spreads and transmission falls due to the build-up of population immunity. As the reviewer has said, this is a matter of notation rather than substance; our definitions are clear (not disputed anyhow), so we have not changed the notation.

===PREPARING YOUR MANUSCRIPT===

- one version identifying all the changes that have been made (for instance, in coloured highlight, in bold text, or tracked changes);
- a 'clean' version of the new manuscript that incorporates the changes made, but does not highlight them. This version will be used for typesetting.

*** The two versions have been prepared

*** Equations are editable

*** Done

*** Done